# Human expansion into Asian highlands in the 21st Century and its effects

Chao Yang [1,2], Huizeng Liu [1,3], Qingquan Li [1] ✉, Xuqing Wang[4], Wei Ma[5], Cuiling Liu[1,2], Xu Fang[1,6], Yuzhi Tang[1], Tiezhu Shi[1,2], Qibiao Wang[7], Yue Xu[1], Jie Zhang[8], Xuecao Li[9], Gang Xu [10], Junyi Chen[11], Mo Su[12], Shuying Wang [1], Jinjing Wu[1], Leping Huang[1], Xue Li[1] & Guofeng Wu [1,2] ✉

Most intensive human activities occur in lowlands. However, sporadic reports indicate that human activities are expanding in some Asian highlands. Here we investigate the expansions of human activities in highlands and their effects over Asia from 2000 to 2020 by combining earth observation data and socioeconomic data. We find that ~23% of human activity expansions occur in Asian highlands and ~76% of these expansions in highlands comes from ecological lands, reaching 95% in Southeast Asia. The expansions of human activities in highlands intensify habitat fragmentation and result in large ecological costs in low and lower-middle income countries, and they also support Asian developments. We estimate that cultivated land net growth in the Asian highlands contributed approximately 54% in preventing the net loss of the total cultivated land. Moreover, the growth of highland artificial surfaces may provide living and working spaces for ~40 million people. Our findings suggest that highland developments hold dual effects and provide new insight for regional sustainable developments.

Covering one-third of the world's land area, Asia comprises numerous highlands, including hills and mountains. The current population of Asia is over 4.5 billion (nearly 60% of the global population), with an increase of 850 million since 2000 (Supplementary Data 1), which has led to the intensification of human activities for meeting the basic life requirements[1–5]. Compared to plains as well as low-lying and non-sloping lands, mountains and hills are generally less economically and environmentally suitable for intensive human activities (such as agriculture, industry, transportation and human settlement)[6,7]. The

geographically diverse highlands are ecologically fragile regions which play important roles in biodiversity conservation (offering habitat for ~33% of terrestrial biodiversity), carbon sequestration, water supply (over 50% of global fresh water) and soil and water conservation[8–12]. Recent studies indicates that human activities are still expanding regionally in some Asian highlands, such as the cultivated land expansions in Southeast Asia[13,14] and the hillside urban land expansions in China[7,15]. Moreover, many of these expansions have caused forest loss[3,13,14,16,17]. These types of regional human activities in highlands

[1]MNR Key Laboratory for Geo-Environmental Monitoring of Great Bay Area & Guangdong-Hong Kong-Macau Joint Laboratory for Smart Cities & Guangdong Key Laboratory of Urban Informatics & Shenzhen Key Laboratory of Spatial Smart Sensing and Services, Shenzhen University, Shenzhen 518060, China. [2]School of Architecture and Urban Planning, Shenzhen University, Shenzhen 518060, China. [3]Institute for Advanced Study, Shenzhen University, Shenzhen 518060, China. [4]Center for Hydrogeology and Environmental Geology, China Geological Survey, Nanjing 210000, China. [5]School of Civil Engineering, Chongqing Jiaotong University, Chongqing 400074, China. [6]College of Electronics and Information Engineering, Shenzhen University, Shenzhen 518060, China. [7]Anhui Zhonghui Urban Planning Survey & Design Institute, Tongling 244000, China. [8]College of Information and Electrical Engineering, China Agricultural University, Beijing 100083, China. [9]College of Land Science and Technology, China Agricultural University, Beijing 100083, China. [10]School of Resource and Environmental Sciences, Wuhan University, Wuhan 430079, China. [11]Key Laboratory of Virtual Geographic Environment of the Ministry of Education, Nanjing Normal University, Nanjing 210000, China. [12]Shenzhen Urban Planning and Land Resource Research Center, Shenzhen 518034, China. ✉e-mail: liqq@szu.edu.cn; guofeng.wu@szu.edu.cn

diverge from the general view that the expanding agricultures or human settlements in hills and mountains are not economically feasible (see the high-resolution evidence of typical human activity expansions in some Asian highlands; Supplementary Figs. 1–5).

The United Nations Food and Agriculture Organization (FAO) and Intergovernmental Panel on Climate Change (IPCC) frequently apply foundational assumptions of various human activity expansions as important input factors for modelling global and regional climate changes. For example, FAO estimated little or no additional net cultivated land expansions in global mountains[18], and this estimation was put into future scenarios for climate change impact assessments[19,20]. However, the evidences obtained in the past decade of the Southeast Asian mountains deviate from this estimation. The current reports on human activity expansions in the Asian highlands are not complete because they only include a small part of human activity expansions with limited geographical coverage and only assess the impacts on forest loss without considering other ecological land types (e.g. grasslands, wetlands and shrublands)[13,14]. A complete assessment of human activity expansions in highlands can improve the inputs for the IPCC climate change model, provide basis for highland ecological conservation and help to understand regional sustainable development processes. Unfortunately, the human efforts on highland developments over Asia are still unknown.

Our study was designed to answer the following questions: (1) how many human activity expansion areas have been increased across Asian highlands since the 21st century, in which countries and to what degree? and (2) what are the visible effects of human activity expansions on highlands? To answer these two questions, we first employed multiple state-of-the-art satellite images (30 m resolution) from 2000 to 2020 to reveal highland human activity expansions (all the expansion areas of cultivated lands and artificial surfaces in hills (elevation 0–1000 m with slope >15°or elevation 1000–1200 m with slope >8°or elevation 1200–1500 with slope 0–3°) and mountains (elevation 1200–1500 with slope >3°or elevation >1500 m)) in 48 Asian countries (see details in Methods part 'Measuring highland human activity expansions'). Then, we applied socioeconomic data to estimate the visible effects of highland human activity expansions, including ecological land loss, habitat fragmentation, contribution rate of highland cultivated land net growth to the total cultivated land conservation and population capacity supported by highland artificial surface expansions (see details in Methods part 'Assessing effects of human activity expansions in highlands'). The validation using approximately 22,000 high-resolution samples (Supplementary Data 2–4; Supplementary Figs. 6 and 7) confirmed the accuracies of human activity expansions in the highlands and ecological land losses (details in last part of Methods). The detection accuracy of human activity expansions in the highlands was more than 90% (Supplementary Table 1). Spatially, the detection accuracy of human activity expansions in the highlands was similar to that in the lowlands (93.13% versus 94.80%; Supplementary Table 1). Moreover, the detection accuracy of ecological land loss induced by highland human activity expansions was around 88%, and the correct detection accuracies of ecological land losses in highlands and lowlands were close (88.92% and 88.45%, respectively; Supplementary Table 2).

## Results

### Human activity growth in the highlands across Asia during 2000–2020

In answering the first question, widespread human activity expansion areas were observed in the Asian highlands over 2000–2020 (Fig. 1), confirming the aspirations of humans for developing the highlands in Asia. Generally, about 23% of the human activity growth areas occurred in highlands (45% in hills and 55% in mountains,

Supplementary Table 3) (Fig. 2a), indicating that humans have developed large mountainous areas. Our results also exhibited the spatial heterogeneity of human activity expansion rates in the highlands in West (32.31%), East (29.23%), Southeast (13.14%), South (9.77%) and Central (4.45%) Asia (Fig. 2a). The dominant human activity expansion type in the Asian highlands was cultivated land (~80%), and artificial land accounted for 20% of the expansions (Fig. 2a). Moreover, the proportion of cultivated land growth in highlands differed in West, East, Southeast, South and Central Asia (77–92%), and that of highland artificial surface expansions varied from 8% to 23% (Fig. 2a). Southeast and South Asia held the largest proportions of cultivated land expansion (more than 91%), while West and East Asia had the largest proportion of artificial surface expansion (>22%, above the Asian level) (Fig. 2a). Overall, human activity expansions in mountains were higher than those in hills for Asia. However, the Southeast Asia is contrary (88% in hills versus 12% in mountains; Supplementary Table 3), indicating its tendency in developing hilly areas. Furthermore, about 27% of the cultivated land expansions and 14% of the artificial surface growths were observed in Asian highlands (Fig. 2b). The expansion rates of cultivated lands and artificial surfaces in West and East Asia highlands were higher than Asia's overall rates (cultivated land expansion rate >16% and artificial surface expansion rate >37%) (Fig. 2b).

The human activity expansions in highlands were observed in all the 48 Asian countries with great spatial heterogeneity during 2000–2020 (Fig.1 and Fig. 2c, d), indicating that highland development is a common phenomenon in Asia. The highland human activity expansion rate in 15 countries exceeded the Asia level (Asia level ≈ 23%), and the top 10 countries with the highest highland human activity expansion rates were Bhutan (89.25%), Nepal (75.23%), Armenia (71.12%), Iran (56.85%), Yemen (55.16%), Afghanistan (53.71%), Kyrgyzstan (50.28%), Mongolia (42.13%), Turkey (42.08%) and North Korea (37.61%) (Fig. 2d). Most of these countries have very high proportion of highlands, in which Yemen only holds 40.35% highlands but its expansion rate in highlands was more than 55% (Fig. 2d), which is an unusual contrary observation in Yemen compared with other countries. Additionally, the proportion of artificial surface expansions exceeded that of cultivated land expansions in the highlands of Maldives, Brunei, Singapore, East Timor, Bahrain, Israel, Jordan, Kuwait, Lebanon, Oman, Palestine, Qatar, Saudi Arabia and United Arab Emirates (Fig. 2e), most of which are high or upper-middle income countries. The proportion of cultivated land expansion in highlands was close to or greater than 90% in North Korea, Cambodia, Indonesia, Laos, Myanmar, Philippines, Thailand, Vietnam, Kazakhstan, Bhutan, India, Pakistan, Nepal, Afghanistan, Cyprus and Yemen (Fig. 2e), most of which are low or lower-middle income countries. Furthermore, the artificial surface expansion rate in the highlands of 14 countries exceeded the Asia level (Asia level ≈27%) (Supplementary Fig. 8a), and the cultivated land expansion rate in the highlands of 16 countries was higher than the Asia level (Asia level ≈14%) (Supplementary Fig. 8b).

### Ecological land loss and fragmentation induced by highland developments

In answering the second question, human activity expansions in highlands caused large ecological land loss in a negative perspective (here ecological land is defined as the sum of forest, grassland, wetland and shrubland) (Supplementary Fig. 9). Overall, 76% of the human activity expansion areas in highlands were changed from the ecological lands in Asia (the proportions in hills and mountains were close, 49% and 51%, respectively Supplementary Table 4) (Fig. 3a), 95% in Southeast Asia and 62–77% in West, East, South and Central Asia (Fig. 3a), indicating that the highland developments in Southeast Asia resulted in greater ecological cost. About 87% of the cultivated land expansions and 35% of the artificial surface expansions in highlands came from ecological lands in Asia (Fig. 3b), indicating the tendency to

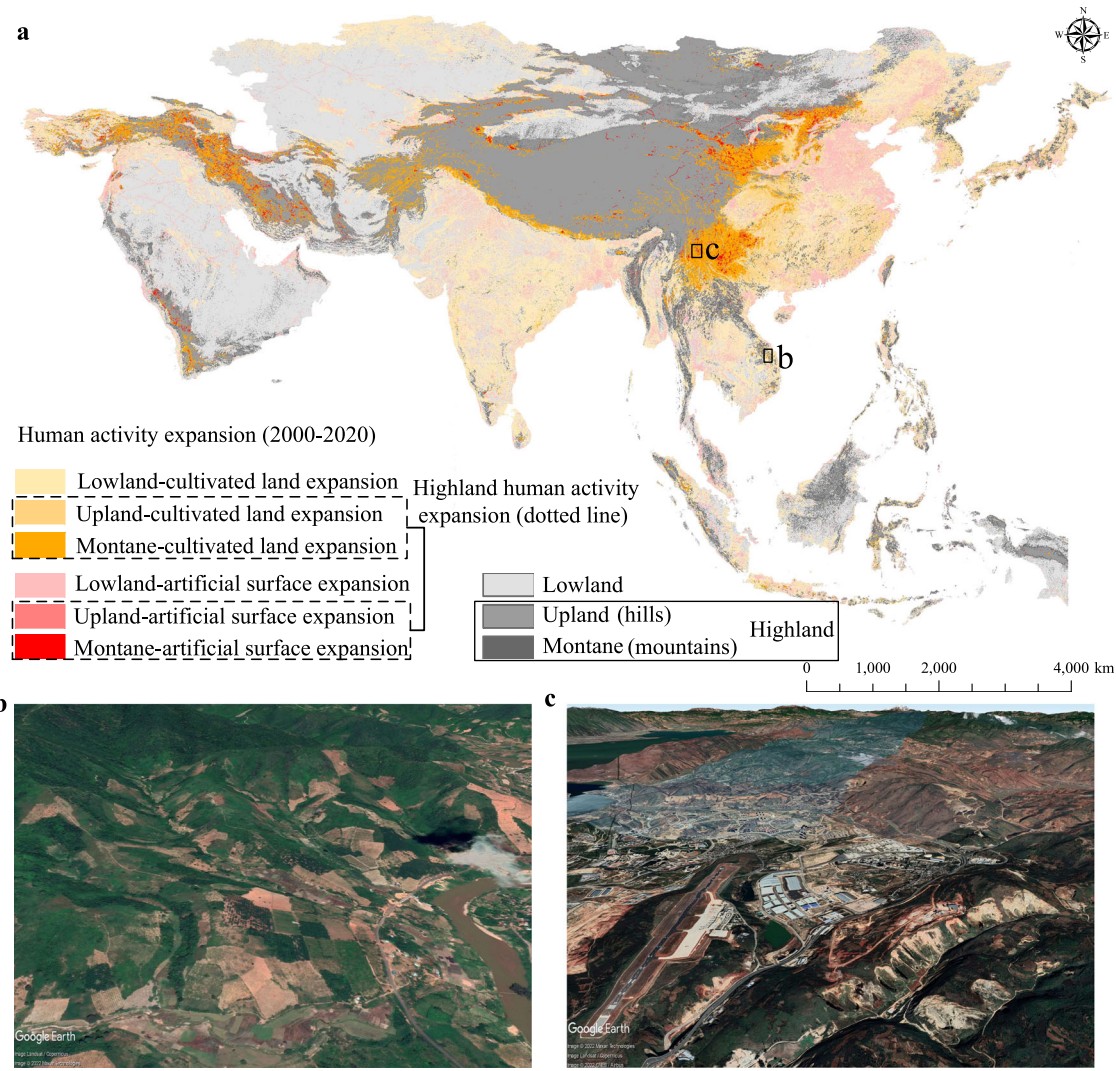

**Fig. 1 | Spatial distribution of human activity expansions in Asia from 2000 to 2020. a** Map of human activity expansion areas according to landforms and land use information (30 m resolution), **b** cultivated land expansion in the highlands obtained from Google Earth Pro (14°41'17.06"N, 107°56' 56.01"E, spatial resolution <5 m), and **c** artificial surface expansion in the highlands obtained from Google Earth Pro (25°39' 45.35"N, 100°19' 26.93"E, spatial resolution <5 m).

develop agriculture in ecological lands and to expand artificial surfaces in non-ecological lands. The rates of ecological lands occupied by cultivated land and artificial surface expansions in highlands in Southeast Asia were higher than those in other Asian regions (Fig. 3b). Moreover, the human activity expansions in the entire Asian highlands were dominated by the encroachments of forests and grasslands (i.e. forest and grassland account for about 45% and 49% of the encroachments, respectively) (Fig. 3d). Central and Southeast Asia held the largest proportion of grasslands (92%) and forests (81%), respectively (Fig. 3d).

The habitat fragmentation analyses of ecological lands in highlands (details in Methods) showed that habitat fragmentation increased from 2000 to 2020 in Asia (Supplementary Fig. 10), indicating that the highland developments intensified habitat fragmentation. Furthermore, the degree of habitat fragmentation in hilly areas (~0.97) was significantly higher than that in mountainous areas (~0.65) (Supplementary Fig. 10), indicating that humans strongly affected the highlands and the habitat fragmentations in ecological lands in hilly areas were serious. However, the increasing tendency of fragmentation for mountains was larger than that for hills from 2000 to 2020 (Supplementary Fig. 10), which is consistent with the increase in human activities in the mountainous area

observed herein (the expansion rate of human activities in Asian mountains was higher than that in hills). The ecological land loss induced by highland developments exhibited spatial heterogeneity at the country level (Fig. 3d). More than 90% of the human activity expansions in highlands were changed from ecological lands in 15 countries, including 10 Southeast Asian countries (Supplementary Fig. 9a), most of which are low or lower-middle income countries; while less than 10% of the human activity expansions were observed in United Arab Emirates, Saudi Arabia, Maldives, Qatar, Kuwait and Bahrain (no highland ecological land loss) (Supplementary Fig. 9a), which are high or upper-middle income countries. The ecological land loss rate in low and lower-middle-income countries was 1.9 times that in high and upper-middle-income countries in Asia (Fig. 3e), indicating the highland developments with higher ecological costs in low and lower-middle-income countries. Moreover, the dominant type of ecological land loss was forest or grassland in 45 countries and shrubs in United Arab Emirates, Saudi Arabia and Yemen (Supplementary Fig. 9b). In more than half of the Asian countries, the ecological land loss rates in highlands caused by the expansions of cultivated lands or artificial surfaces exceeded the Asia level (Supplementary Fig. 11), indicating the considerable ecological effects of highland developments in these countries.

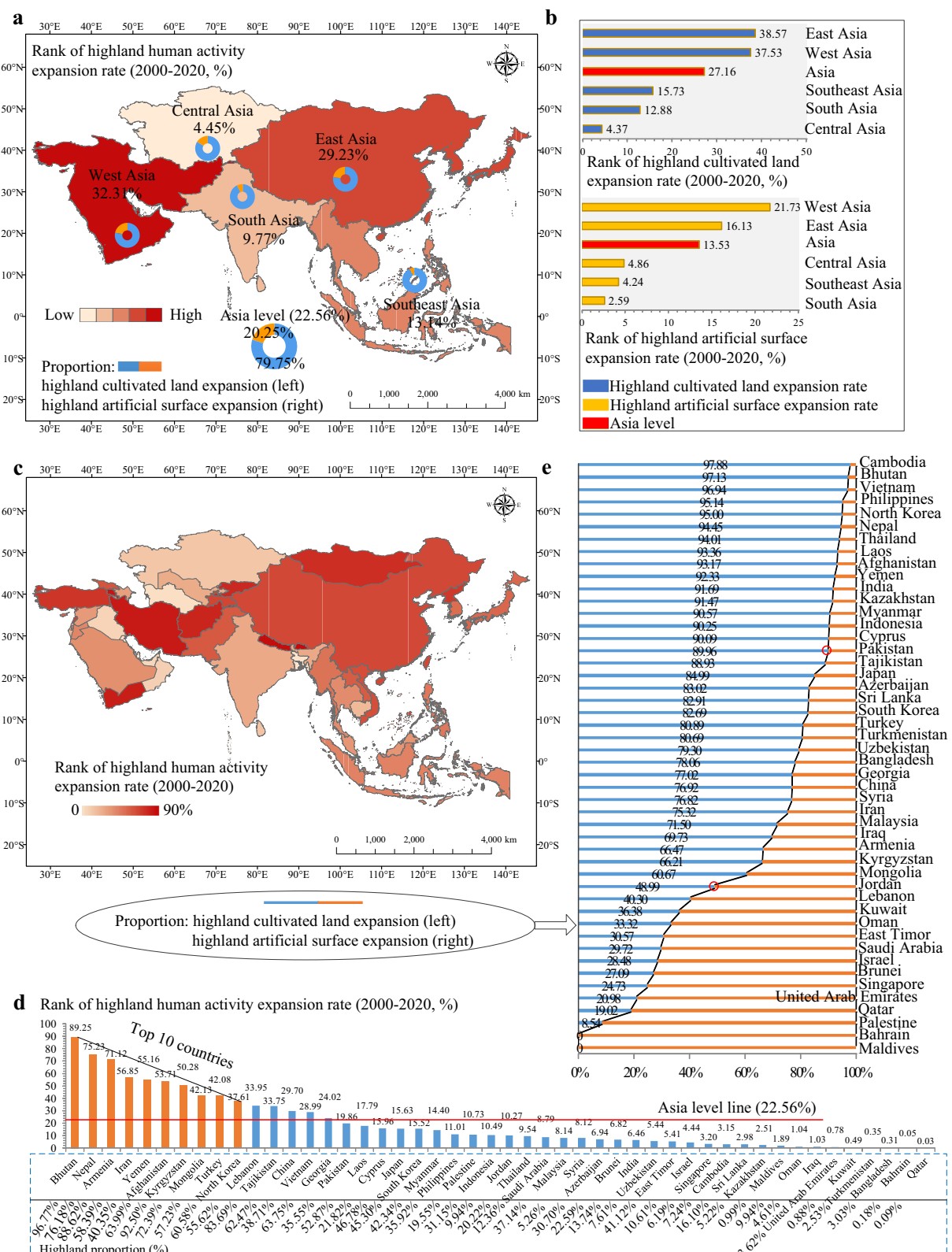

**Fig. 2 | Quantification of human activity expansions in Asian highlands from 2000 to 2020. a** Rank of human activity expansion rate, **b** rank of human activity expansion rates considering cultivated land and artificial surface types, **c** spatial variability of human activity expansion rate, **d** rank of human activity expansion rate in 48 Asian countries and **e** proportion of human activity expansion types in 48 Asian countries.

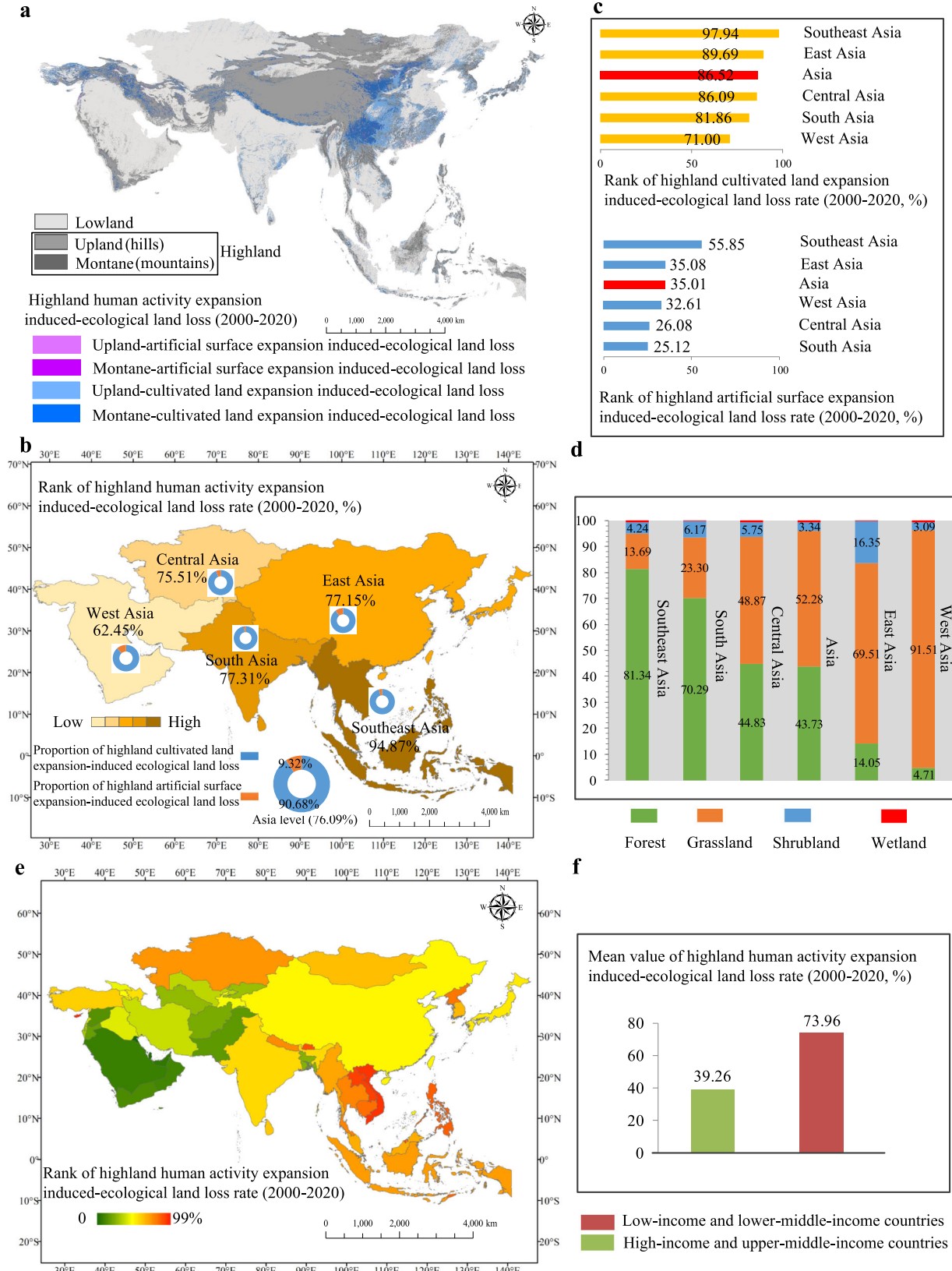

**Fig. 3 | Quantification of ecological land loss in highlands induced by human activity expansions in Asia from 2000 to 2020. a** Spatial distribution of ecological land loss induced by human activity expansions, **b** rank of human activity expansion rate, **c** rank of human activity expansion rates considering cultivated land and artificial surface types, **d** spatial variability of human activity expansion rate, **e** proportion of ecological land loss types caused by human activity expansions, and **f** mean rate value of ecological land loss induced by human activity expansions.

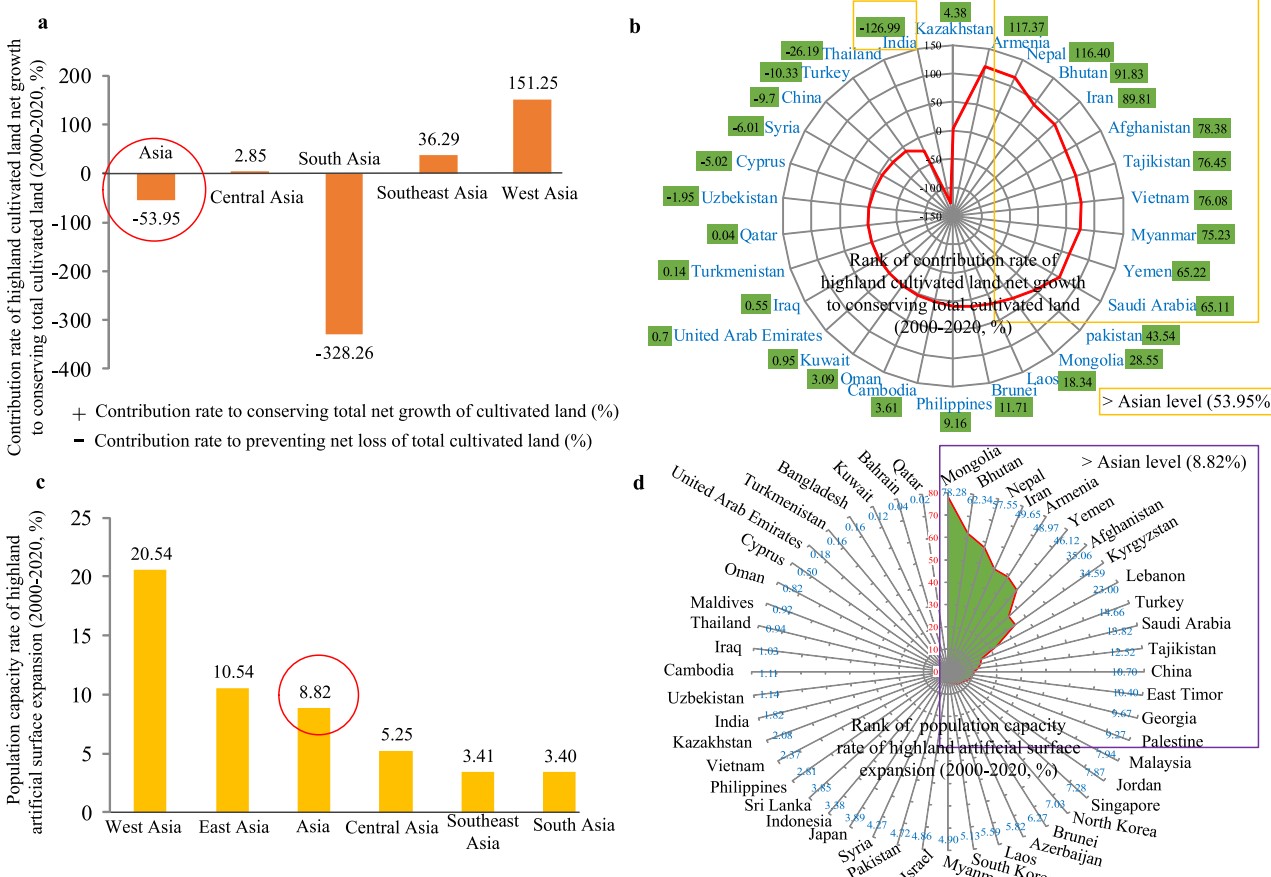

**Fig. 4 | Highland developments supporting the developments in Asia from 2000 to 2020. a** Contribution rates of cultivated land net growth in highlands for conserving total cultivated lands, **b** rank of contribution rates of cultivated land net growth in highlands for conserving total cultivated land, with 30 countries exhibiting net increase of highland cultivated land (negative value (−) denotes the contribution rate of cultivated land net growth in highlands for preventing total net loss of cultivated land; positive value (+) denotes the contribution rate of cultivated land net growth in highlands for conserving total net growth of cultivated land, see details in Methods part), **c** population capacity rates of artificial surface expansions in highlands, and **d** rank of population capacity rates of artificial surface expansions in highlands.

## Highland developments support the developments in Asia

In answering the second question, a net loss of around $3.64 \times 10^4$ km² of cultivated lands was observed in Asia from 2000 to 2020; however, the cultivated land in highlands exhibited a net growth of about $1.96 \times 10^4$ km² in a positive perspective (Supplementary Table 5). The cultivated land net growth in the highlands contributed ∼54% in preventing the net loss of the total cultivated land in Asia (Fig. 4a), and the main contributions came from West and Southeast Asia (sum of net cultivated land growth in highlands was >$1.5 \times 10^4$ km², accounting for approximately 80% of the total contribution rate in Asia). The cultivated land net growth in the highlands in South Asia contributed about 328% in conserving the total cultivated land growth, which is significantly higher than that in other Asian regions (South Asia > West Asia > Southeast Asia ≫ Central Asia) (Supplementary Table 5). East Asia did not contribute at all due to its negative cultivated land growth in the highlands (Supplementary Table 5). The net increase of cultivated lands in the highlands was observed in 30 countries (Fig. 4b), six of which (i.e. China, Vietnam, Afghanistan, Iran, Yemen and Nepal) had a net growth area of more than 1500 km² (total contribution rate reached 99% for overall Asia) (Supplementary Table 6). The contribution rates of 11 countries in conserving the total cultivated land exceeded the Asia level (Asia level ≈54%) (Fig. 4b, 11 countries inside the yellow border), indicating that cultivated land growth in the highlands was the most important way to maintain the stability of cultivated land in these countries.

To address the human living and working space afforded by artificial surface expansion areas in Asian highlands during 2000–2020, the population capacity of artificial surface expansion areas in Asian highlands was calculated using the high-resolution population density data (details in Methods). Our results showed that the artificial surface expansion areas during 2000–2020 in Asian highlands (∼14%) provided living and working space for ∼40 million people (∼39% in hills and ∼61% in mountains) (Supplementary Table 7), indicating that the artificial surface expansion in Asian highlands might somewhat solve the population resettlement problem considering ∼406 million people in lowland artificial surface expansions (Supplementary Table 7). Therefore, the population capacity rate in the highlands accounted for nearly 9% of the total population capacity in Asia (Fig. 4c). Notably, the population capacity rate in artificial surface expansion areas in the highlands of West Asia was close to 21% of the total population capacity, which is significantly more than that in the highlands of East Asia (11%) and far beyond that in the highlands of other regions (Fig. 4c). The population capacity rate of artificial surface expansion areas in the highlands of 16 Asian countries exceeded the Asia level (Fig. 4d, 16 countries inside the purple border), indicating that the contributions of using highlands to carry population in these countries was significantly higher than those in other countries. Notably, among these 16 countries, the highland population capacity rate of Mongolia, Bhutan, Nepal and Iran varied from 50% to 78% (Fig. 4d and Supplementary Table 8), indicating that their artificial

surface developments in the highlands were the dominant method for improving population capacity, considering these 4 countries all have very high highland proportions and highland development rates (Fig. 2d and Supplementary Fig. 8a).

## Discussion

Our study revealed that earth observation-derived human activity expansions in highlands were widely distributed with great spatial heterogeneity and had dual effects in Asia. Moreover, they gave new insights for regional sustainable developments and provided timely, transparent and consistent information on highland developments across Asia. Our results hold several benefits: (1) assisting in validating official statistics and increasing transparency in tracking the spatio-temporal characteristics of human land use management, (2) providing improved imputing for FAO and IPCC climate change models for better climate change understanding, (3) providing information on regulating the over exploitation in highlands and assessing the potential highland developments, and (4) providing a basis for the ecological and biodiversity conservation in highlands.

We analysed the dominant and potential drivers affecting the human activity expansions in the highlands in different Asian countries during 2000 to 2020 from multifactor perspective (Supplementary Table 9). Generally, for the countries with extremely high (≈90%) or very low human activity expansion rates in the highlands (<2%) (only one-fifth of countries), topographic constraint may be the dominant factor affecting highland developments, considering that these countries possess extremely high or low proportion of highlands (Supplementary Table 9). However, other four-fifths Asian countries had 5–95% highland areas (Supplementary Fig 12), and thus explaining the human activity expansion rates in the highlands in these countries using topographic factors is unfair. Highland developments may be driven by different needs, and the superposition of multiple factors could result in the highland developments with different degrees in these countries, including topographic condition, marketing demand, population growth/livelihood, and dynamic socio-political economy/ecology factors (details in Supplementary Table 9). Highland developments require economic support (especially for artificial surface expansions[21]). Therefore, this study also summarised the general rules of highland developments according to the human activity expansion types in highlands and the economic levels of different countries, i.e. high or upper-middle income countries tended to develop artificial surfaces in the highlands, and low, or lower-middle-income countries tended to develop cultivated lands in the highlands (significantly correlated to the economic level, $P < 0.01$) (Supplementary Fig. 13).

We observed that highland developments caused considerable ecological land loss and fragmentation. Transforming ecological lands into human use in highlands exhibited higher ecological costs in low and lower-middle-income countries (~1.9 times the rate of ecological land loss) than in high and upper-middle-income countries, which corresponds to the law of the environmental Kurtz curve in the process of economic development in developing and developed countries. Previous studies have proved that, for countries just in the stage of economic take-off (e.g. low and lower-middle income countries), reducing poverty and promoting rapid economic growth force these countries to make extensive use of natural endowment to realise industrial take-off, resulting in ecological damage in the early stage of development[22–24]. In addition, international trade and investment can lead to the situation of "producing products and consuming natural resources in low-income countries, while enjoy benefits in high-income countries", which improves the ecological environments of developed countries and exacerbates the ecological problems of developing countries[25]. Therefore, the above evidences may well explain the large ecological costs caused by the highland developments in low and low-income countries. Notably, the increasing cultivated land and artificial surface patches may erode and divide the landscape formed by

ecological lands[26,27], causing habitat fragmentation of the ecological lands in the highlands. As a result, the extinction risks of endemic species will increase, the highland regions will become warmer[2,6,28,29] and frequent human–animal conflicts will occur[30]. For example, frequent human–animal conflicts (e.g. human conflicts with elephants, bears and monkeys) were observed[30–32] mainly due to the replacements of highland habitats by cultivated lands and artificial surfaces. A recent study suggested that mountain deforestation induced warming and local temperature anomaly (up to 2°C)[6]. Thus, more obvious warming may be observed in the regions with high forest occupation proportions (Southeast and South Asia) when developing highlands. Generally, serious ecological land loss and fragmentation in highlands considerably impact biodiversity, carbon sequestration, soil erosion, landslides and water regulation, and thus, Asian countries should consider ecological protection when developing highlands. Notably, lowland population is increasingly dependent on mountain water resources supply[14]; however, the cultivated land expansions in highlands force farmers to input more fertiliser and chemical pesticides, which may decrease the water quality of mountains. Therefore, mountainous, low and lower-middle-income countries should accelerate economic transformation, reduce over-reliance on agriculture or provide alternatives.

Although Asian highland developments prevented the net loss of total cultivated lands and provided living and working spaces for the growing population, considering the negative ecological impacts of highland developments, Asian countries should balance highland developments and ecological protection for their sustainable developments rather than blind developing. Previous studies have proved that "land sharing" (i.e. integrating biodiversity conservation and food production on the same land with wildlife-friendly farming methods) and "land sparing" (i.e. separating land for conservation from land for crops, with high-yield farming facilitating the protection of remaining natural habitats from agricultural expansion) could provide alternatives on supporting agricultural sustainable developments while mitigating negative effects[33–35]. Meanwhile, "getting road expansion on right track" (i.e. improving and planning transport links to substantially increase food production at relatively limited environmental cost) could enhance livelihoods and regional food production and reduce overexploitation of cultivated land while helping safeguard vital ecosystem services and significant biological diversity[36]. Therefore, integrating the strategies of "land sharing", "land sparing" and planning transport links in Asian highland developments may help save land resources and protect the ecological environment in highlands, and make highland developments achieve sustainability.

To our knowledge, this study was the first to reveal human activity expansions and their effects in Asian highlands; however, some uncertainties and future works still need to be further explored. For uncertainties, the 30 m spatial resolution of the images makes the detections of accurate human activity expansion boundaries in highlands difficult. However, the combination of the National Aeronautics and Space Administration (NASA) Landsat satellite with the European Space Agency (ESA) Sentinels satellite[37] may increase the data available. The accuracy of the population capacity of artificial surface expansion in the highlands depends on population density, which relies on the accuracy of satellite-based settlement and building mapping. In the regions where settlements/buildings are missed in the population density data production process, the population may be overallocated to neighbouring settlements. In contrast, in the regions where settlements/buildings are incorrectly identified, the population may be under-allocated to neighbouring settlements. Since no unified standard for the division of highlands exists, we used the latest standards (Supplementary Table 9); however, other standards (e.g. World Conservation Monitoring Centre-UN Environment Program (WCMC-UNEP)[38]) may also hold potentials in mapping Asian highlands. Although Asian

highlands hold great potentials for social developments, we should explore the threshold of balancing highland developments and ecological protection in future works. Moreover, considering that highlands are rich in biodiversity, we need to further explore the impacts of highland developments on biodiversity.

In summary, our results revealed the widespread human activity expansions in highlands over Asia: ~23% of human activity growth areas were present in highlands, while most expansion areas came from ecological lands and intensified habitat fragmentation, with potential implications of biodiversity loss, warming, water security, and ecosystem service degradation. Moreover, the highland developments resulted in large ecological costs in low or lower-middle-income Asian countries. Parallel initiatives for highland developments and governance as well as accelerating economic transformation may alleviate or prevent large ecological costs. Integrating the most accurate information of Asian highland developments into climate models to obtain more accurate prediction of global climate change is ugent[19], and our results may contribute greatly to climate change modelling. In addition, Asian highlands as a hotspot of biodiversity due to their large-scale protected areas and threatened species[39], the future human activity expansions in the highlands driven ecological land loss should warrant the attention of governments and policymakers.

## Methods
First, we detected and quantified human activity expansions in highlands (mainly referring to the expansions of cultivated lands and artificial surfaces in highlands) from 2000 to 2020 in 48 Asian countries, including East Asia (China, Mongolia, North Korea, South Korea and Japan), Southeast Asia (Philippines, Vietnam, Laos, Cambodia, Myanmar, Thailand, Malaysia, Brunei, Singapore, Indonesia and East Timor), South Asia (Nepal, Bhutan, Bangladesh, India, Pakistan, Sri Lanka and Maldives), Central Asia (Kazakhstan, Kyrgyzstan, Tajikistan, Uzbekistan and Turkmenistan) and West Asia (Afghanistan, Iraq, Iran, Syria, Jordan, Lebanon, Israel, Palestine, Saudi Arabia, Bahrain, Qatar, Kuwait, United Arab Emirates, Oman, Yemen, Georgia, Armenia, Azerbaijan, Turkey and Cyprus). We then investigated the visible effects of human activity expansions in the highlands (mainly referring to the ecological land loss and habitat fragmentation, the contribution rate of cultivated land net growth and population capacity of artificial surface expansion in highlands).

### Digital elevation model (DEM)-derived landforms
Freely available Advanced Spaceborne Thermal Emission and Reflection Radiometer Global DEM (ASTER GDEM) products with WGS-84 coordinate system and 30 m spatial resolution, which were captured around or after 2000 in Asia, were downloaded from the NASA website (http://reverb.echo.nasa.gov/reverb/). ASTER GDEM was jointly developed by the Ministry of Economy Trade and Industry (METI) of Japan and NASA of the USA by stacking all individual cloud- and non-cloud-masked scene DEMs while applying various algorithms to remove abnormal data[40]. ASTER GDEM products have been widely used in various fields, and their reliability have been proven[41]. We used ASTER GDEM (3935 tiles) to derive the major landforms (i.e. lowland and highland)[42] in Asia as the base map for subsequent analyses. Lowland includes plains and terraces, while highland refers to the collection of upland (hills) and montane (mountains). The classification criteria of highland and lowland are based on elevation and slope (see details in Supplementary Table 10), which is summarised by Margono and colleagues according to large previous studies[42]. The spatial distributions of lowlands and highlands in Asia were identified with the batch processing module of ArcGIS 10.2 platform. Overall, highlands and lowlands are widely distributed in Asia, and their areas accounted for about 40.41% and 59.59%, respectively (Fig. 1).

### Satellite-based high-resolution land cover products
Some freely available global land cover products with fine spatial resolutions (≤500 m) exist[43,44], including the MODIS Land Cover Type (MLCT) series products from 2001 to 2016 (500 m), ESA Climate Change Initiative (ESA-CCI) land cover product from 1992 to 2015 (300 m), Finer Resolution Observation and Monitoring of Global Land Cover product (FROM-GLC) in 2010, 2015 and 2017 (30 m and 10 m), global 30 m land cover classification with a fine classification system (GLC_FCS30) in 2015 and 2020 (30 m) and global land cover (Globeland30) data product in 2000, 2010 and 2020 (30 m). Since human activities need to be detected in detail (such as the activities in small hilly areas), we tested all these products and found that the minimum geographical unit (a pixel) of MLCT ($500 \times 500$ m$^2$) and ESA-CCI ($300 \times 300$ m$^2$) could not completely detect some small hills, which may underestimate human activity areas. Moreover, FROM-GLC and GLC_FCS30 are limited by historical archives and are thus not suitable for studying long-term land cover change. Thus, we selected Globeland30 as the land cover data to quantify the human activity expansions in the highlands. Globeland30 datasets (http://www.globallandcover.com/) have the same coordinate and projection systems as DEM, and they were developed by combining several multispectral images, including Landsat Thematic Mapper (TM5) and Enhanced Thematic Mapper Plus (ETM + ), Chinese HJ-1 images and 16 m Gaofen (GF-1) multispectral images (2020 version). The classification was performed using a split-and-merge strategy, Pixel–Object–Knowledge approach was applied to classify each land cover type and a knowledge-based interactive verification procedure was applied to improve mapping accuracy[45]. Globeland30 datasets cover 10 land cover types: cultivated land, artificial surface, forest, grassland, shrubland, wetland, water, tundra, bare land, glaciers and permanent snow. The accuracy of Globeland30 was assessed using the landscape shape index-based sampling model and more than 230 thousand samples, and the overall accuracy and kappa coefficient, respectively, were 85.72% and 0.82 for the 2020 version and about 83.50% and 0.78 for the previous versions[45,46], which meets the accuracy requirement of land cover change analysis[47]. Consequently, the Globeland30 products have been extensively used for a variety of applications[48,49].

### Socio-economic datasets
Socioeconomic datasets include population, economic level and high-resolution population density for all 48 Asian counties. Population and economic levels were collected from the World Bank database (https://data.worldbank.org.cn/) (see Supplementary Data 1). The World Bank divides the economic levels of different countries into low, lower-middle, upper-middle and high income based on the gross national income per capita. According to this criterion, Asia covers 12 high, 15 upper-middle, 16 lower-middle and 5 low-income countries (Supplementary Data 1). The high-resolution population density gridded dataset from 2020 with 100 m resolution was obtained from WorldPop (www.worldpop.org/project/categories?id=3). This dataset is currently the most accurate estimation of population density, and it includes constrained and unconstrained type data. In this study, we employed constrained type data as they were developed using a top-down constrained method and Random-Forests-based dasymetric redistribution (i.e. the output yields a more accurate population distribution without the prediction of small population numbers in likely uninhabited areas as it employs building footprints and/or built settlements as inputs)[50,51]. In contrast to unconstrained type data, the residential populations of small settlements and isolated buildings were identified in constrained type data[50]. WorldPop's population density dataset holds the same coordinate system as DEM and Globeland30, and its unit is the number of people per pixel with country totals adjusted to match the corresponding official report of the Population Division of the Department of Economic and Social Affairs

of the UN Secretariat. We applied the nearest neighbourhood algorithm to resample the 100 m resolution population density dataset to 30 m to match the spatial resolution of DEM and GlobeLand30.

## Administrative boundary

The administrative boundary data of the 48 Asian countries were obtained from the Database of Global Administrative Areas (GADM) (https://gadm.org/) hosted by the University of California at Davis. The GADM data provide high-resolution shape files at administrative levels, including country and state or provincial level. The data with the latest version (v.3.6) were used in our study.

## Measuring highland human activity expansions

We selected the expansions of cultivated lands and artificial surfaces as the representatives of human activity expansions. According to the classification system of GlobeLand30, cultivated land denotes the land used for planting crops, including paddy fields, irrigated dry land, rainfed dry land, vegetable land, pasture land, greenhouse land and land mainly for planting crops with fruit trees and other economic trees and economic crops, such as tea and coffee gardens[46]. Artificial surface is impervious surface formed by artificial construction activities, including various residential settlements, such as towns, industrial and mining areas and transportation facilities, but excluding contiguous green spaces and water bodies within construction lands[46]. Therefore, based on the above definitions, the expansions of cultivated lands and artificial surfaces cover almost all intensive human activities that may occur in the highlands. To improve computational efficiency, the following strategies were applied. The Python platform with a target-level change detection method was applied to detect the change regions (i.e. changed pixels) of cultivated lands and artificial surfaces in Asia from 2000 to 2020, including increased (i.e. expanded areas of human activities), unchanged and decreased areas. The human activity expansions in lowlands and highlands were then classified and quantified according to landform information and change detection results. The human activity expansions in the highlands include the expansions of cultivated lands and artificial surfaces in the upland and montane (Fig. 1). Finally, we compared the rates of human activity expansions in the Asian highlands (i.e. highland/(lowland + highland) × 100%).

## Assessing the effects of human activity expansions in highlands

This study quantified four human activity expansion effects in Asian highlands: ecological land loss (i.e. forest, grassland, wetland and shrub, which serve as fundamental ecological lands[52]), habitat fragmentation of ecological land, the contribution rate of cultivated land net growth in highland to the total cultivated land conservation and population capacity of artificial surface expansion. The ecological land loss in highlands induced by human activity expansions denotes the areas of ecological lands in highlands at time T that are transformed into cultivated land/artificial surfaces at T + 1. A transfer matrix with Python and ArcGIS was developed to detect the ecological land loss in the highlands. To calculate the habitat fragmentation degrees of ecological lands in the highlands caused by human activity expansions, a habitat fragmentation index (HFI) (Eq. 1) was employed. HFI can reflect the fragmentation degrees of habitat areas from a patch perspective. The HFI value varies from 0 to 1, 0 and 1 imply no and complete habitat fragmentation, respectively, and a value closer to 1 indicates a high habitat fragmentation and a strong impact of human activities[53]. The HFI values from 2000 to 2020 were calculated with Python and ArcGIS (Supplementary Fig. 10).

$$HFI_i = 1 - \max(A_{ij}) / \sum_{j=1}^{N_i} A_{ij} \qquad (1)$$

where $HFI_i$ denotes the habitat fragmentation degree of landscape class $i$ ($i$ refers to the ecological land in the highlands), $A_{ij}$ is the area of

patch $j$ of landscape class $i$ and $N_i$ denotes the total patch mumble of landscape class $i$.

The contribution rate of highland cultivated land net growth to the total cultivated land conservation and population capacity of artificial surface expansion in the highlands were estimated using Eqs. 2 and 3, respectively. The contribution rate of the cultivated land net growth in the highlands for conserving the total net increase of cultivated lands was calculated by dividing the net growth area of highland cultivated land by the total net growth area of cultivated land (Eq. 2; Supplementary Tables 5 and 6). The population capacity of artificial surface expansions in the highlands was estimated by individually matching the population density pixels with artificial surface expansion area pixels (Eq. 3; Supplementary Tables 7 and 8).

$$Cr = (H_{T+1} - H_T) / ((H+L)_{T+1} - (H+L)_T) \qquad (2)$$

$$Pc = w \sum_{i=1}^{n} (N1 + N2 + N3 + \cdots Nn) \qquad (3)$$

where $Cr$ is the contribution rate of cultivated land net growth in the highlands for conserving the total net growth of cultivated lands (a net growth of cultivated land occurs in the highlands and the total cultivated land is a net growth) or the contribution rate of cultivated land net growth in the highlands for preventing the net loss of total cultivated lands (a net growth of cultivated land occurs in highlands and the total cultivated land growth is negative); $H_{T+1}$ and $H_T$ represent the cultivated land area in highlands at time T + 1 and T, respectively; $L_{T+1}$ and $L_T$ represent the cultivated land area in lowlands at time T + 1 and T, respectively. $Pc$ is the total population capacity; $w$ represents the sampling coefficient of population density data; $n$ is the class of the population density pixel, and $N$ is the total number of pixels with different population density classes. Since the population density with 100 m resolution was sampled to 30 m, the sampling coefficient was 0.09 ($w = 0.09$) in this study.

## Accuracy evaluation and analysis

A strategy was designed in our study to verify the result reliability. To assess human activity expansion results, we randomly selected 4008 lowland and 6676 highland (3420 in hills and 3256 in mountains) samples using random function (the sampling interval was a $5 \times 5$ km$^2$ grid) from the human activity expansion areas in Asia (cultivated and artificial land expansions each accounted for about 50% of the sample size) (Supplementary Data 2 and 3; Supplementary Fig. 6). These samples were evenly distributed across the 48 Asian countries. All samples were tagged geographically with Keyhole Markup Language and placed into Google Earth, and the available historical high-resolution images of Google Earth, Landsat TM5 and ETM + satellite imagery from 2000 (around 2000 and cloud-free) were applied to interpret land cover types at all sample sites (total 10,684 samples; Supplementary Data 2 and 3). The collected human activity expansion samples using cloudless high-resolution (≤5 m) imageries around or after 2020 available in the Google Earth, Planet (www.planet.com), in situ measurements (Geo-Wiki, OpenStreetMap, Tencent and Baidu StreetMap) and other public geographic data (Geospatial Data Cloud) were examined to visually interpret whether each sample site was a cultivated land or an artificial surface (Supplementary Data 2 and 3). The visual interpretations of images were performed by a team with specialised knowledge and training in standard procedures and risk assessment. The samples were labelled as human activity expansions if they were a cultivated land or an artificial surface in 2020 but not in 2000 (i.e. correct results); otherwise, they were labelled as non-human activity expansions (e.g. human activity shrinks, stable human activities and stable non-human activities) (i.e. incorrect results) (Supplementary Data 2 and 3). The extremely high resolution satellite images provided by the above platforms include EarlyBird-1 (0.8–3.0 m),

IKONOS (3.2 m), QuickBird (0.6–2.4 m), GeoEye-1 (0.41–1.65 m), WorldView (0.25–1.84 m), Pleiades-1A (0.5–2.0 m), Pleiades-1B (0.5–2.0 m), SPOT-6 and SPOT-7 (1.5–6.0 m), RapidEye (5 m), Doves (3 m; 4-band PlanetScope Scene), Gaofen-1 (2 m/8 m panchromatic band or multispectral bands), Chinese Gaofen-2(1–4 m), Chinese ZY1-02C HRC (2.36 m) and ZY-3 (2.1–5.8 m).

We repeated the sample selection of ecological land loss areas induced by human activity expansions, and finally 4474 lowland and 6275 highland samples of ecological land loss areas (the sampling interval was a $5 \times 5$ km$^2$ grid; hills and mountains each held about 50% of the entire sample size) were randomly selected across Asia (the samples were evenly distributed in forest, grassland, wetland and shrubland loss areas) (Supplementary Data 4 and 5; Supplementary Fig. 7). To each sample, we checked all the cloudless historical high-resolution images collected in 2000 in Google Earth, Landsat TM5 and ETM + satellite imagery to determine whether the sample site comprised forests, grasslands, wetlands or shrublands (total 10,749 samples; Supplementary Data 4 and 5). Then, we examined cloudless high-resolution images collected around or after 2020 from Google Earth, Planet, in situ measurements and public geographic data to visually determine whether the sample sites comprised cultivated lands or artificial surfaces (Supplementary Data 4 and 5). A sample was labelled as ecological land loss induced by human activity expansion if it was an ecological land (i.e. forest, grassland, wetland or shrub) in 2000 but a cultivated land or an artificial surface around 2020 (i.e. correct results); otherwise, the sample was labelled as the non-ecological land loss or ecological land loss induced by non-human activity expansion (i.e. incorrect results) (Supplementary Data 4 and 5). Finally, the number of valid samples were accounted (the samples correctly identified after excluding uncertain or unknown samples), and the overall accuracy metrics[54] (Eq. 4) were calculated to assess detection accuracy (human activity expansion in the highlands and lowlands in Supplementary Table 1; human activity expansion-induced ecological land loss in highlands and lowlands in Supplementary Table 2).

$$OA = \sum_{i=1}^{n} P_i / P \qquad (4)$$

where $OA$ is overall accuracy (i.e. the accuracy of correct detection), $P_i$ represents the samples whose validation results are consistent with classification results (i.e. the correct samples) and $P$ is the total number of samples.

## Data availability

The DEM dataset used in this study are available from NASA (https://earthexplorer.usgs.gov/). The high-resolution land cover products (Globeland30) are available from National Geomatics Center of China (http://www.globallandcover.com/). Population and economic levels are available from the World Bank Database (https://data.worldbank.org.cn/) or from the collated document (Supplementary Data 1). High-resolution population density data (constrained type) are available from WorldPop (www.worldpop.org/project/categories?id=3). Asian country administrative boundaries are available from the GADM (https://gadm.org/). The validation dataset is available from documents (Supplementary Data 2–5). All result data are available in the main text and Supplementary Information.

## Code availability

The programs used to generate all the results were ArcGIS (10.4), Python script, R script, and Excel (2010). Analysis scripts are available on request from the corresponding author.

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

## Acknowledgements

This study was supported by the Basic research project of the Shenzhen Science and Technology Innovation Committee (No. JCYJ20180507182022554), the China Postdoctoral Science Foundation (2021M702233 and 2021M702231) and the National Natural Science Foundation of China (No. 71961137003 and 41890854). We thank all the institutions and platforms for providing supports. We also thank many students and colleagues for their detailed validation works.

## Author contributions

C.Y. and Q.L. designed the study; C.Y., H.L., X.W., C. L. and Y.T. conducted data analyses; C.Y. Q.L. and G.W. prepared manuscript; C.Y., X.W., Y.X., J.W., L.H., X.L. and S.W. conducted sample collection and interpretation work; C.Y., W.M., XC.L., X.F., Y.T., T.S., Q.W., J.Z., G.X., J.C. and M. S contributed to the results.

## Competing interests

The authors declare no competing interests.
