## [Peer Review File · Nature Communications]

Human expansion into Asian highlands in the 21st Century and its effectsEditorial Note: Parts of this Peer Review File have been redacted as indicated to remove third-party material where no permission to publish could be obtained.

REVIEWER COMMENTS

Reviewer #1 (Remarks to the Author):

1. The authors have identified a valid trend in increasing human activity in highlands, this is interesting and significant since the human pressure on highlands can have different and serious impacts on these ecosystems while also contributing to human development. However, it is noteworthy that these trends are not universal as in the case of the Hindu Kush Himalayan region where the population movement to the low lands is still very significant (e.g. as in Ref: P 31 in The Hindu Kush Himalaya Assessment: Mountains, climate change, sustainability and people).
2. To this effect, the questions raised by this paper are noteworthy and contributes significantly to the evidence necessary to account for our response to climate change.
2. The authors have provided sufficient evidence through spatial approaches to answer the questions they raised. The methodology is accessible and easily predictable given the data is shared along with the paper.
3. However, it is not clear what is driving these trends especially when developing highlands is still more costly today than ever given many countries are making it difficult to disturb mountain slopes and mountain ecosystems. There is a need to provide a robust argument for this trend.
4. Though sounds trivial, the authors need to clarify the distinction between hills and mountains (the criteria), especially the criteria used to demarcate/identify them in the spatial datasets they used.
5. Line 116-119: Many of these countries may have the least amount of low land to expand into and are largely hilly in nature and hence the expansion in the highlands is the only natural. There appear to be no interesting or contrary observations to be made here? If some of these countries have plenty of flat lands to expand into, why are they not doing it? It would be interesting to see some arguments on these lines.
6. The paper is good in identifying what is happening but lacks strength in providing an argument on why it is happening. Is it the saturation of lowlands for further expansion, is it the increasing economic and policy ability to occupy highlands more than before? Obviously, these questions are beyond the scope of this paper but are very important offshoots of the questions the authors raised.
7. Though most observations are obvious, the paper identifies the pertinent socio-economic and environmental implications of human expansion into highlands. These observations have important policy implications.
8. Line 157-177: Interesting trends between developing and developed countries in terms of ecological lands being converted to human use. Why developing countries are increasingly relying upon highlands? What is driving them? Policy failures or a conscious decision to engage natural resources for livelihood generation which appears to be the case based on the arguments made in the section starting from L186 but needs to be clearly stated. Do we have evidence supporting these cause-effect relations?
9. It is good that some reasons for expansion are discussed in L 209-212. We need more such possible reasons to make this paper more usable for policy purposes and by establishing cause-effect relations.
10. L 219-224: The question here is why? What is driving these seemingly contrary trends?
11. The text from Line 243 provides some good explanations for trends but not for all the regions, it would be good to provide a clear explanation for all the trends in a tabular form, for e.g.

Reviewer #2 (Remarks to the Author):

Mountains provide 50% of global fresh water, offer habitat for some 33% of terrestrial biodiversity, harbor half of all global biodiversity hotspots (Rumbine & Xu 2021, Front Ecol Environ). Asian highlands are the most dynamic, diverse and complex landscapes at worldwide. This paper provides timely insights for land expansion in the Asian highlands. However it needs to well definitely defined highland boundary. The commonly used definition is from WCMC-UNEP 2002 and Kapos et al. 2000. According to their definition, only around 1/4 of world terrestrial ecosystem is called mountains or hills. It would be nice to tell the difference between hills and mountains for readers.

I like to the statement of "Highland development supports sustainable development in Asia", but authors have failed to provide alternatives on how to do it. Balmford et al. (2016) talked "Getting Road Expansion on Right Track" as a good example. His team is also developed "land sharing" and "land sparing" approach for agricultural expansion.

For drivers of land expansion in highlands, instead of sterotype socioeconomic data, the paper could be benefit from more dynamic political economy/ecology analysis, such as environmental Kuznets Curve for low income and middle-high income countries (Rudel at al. Global Environmental Change 15(1):23-21), transitional economy for Central Asian countries and North Korea, civil wars for Yeman and Afghanistan. The English needs significant improvement too. More specially:

- 1) Fig 2(d) top 10 countries are all mountainous countries, please try to put percentage of mountainous areas for each country;
- 2) Line 585: "Tailand" should be "Thailand"

**Assessing human efforts on highland developments over Asia in the
21st century**

NCOMMS-21-38202-T Response to Reviewers

Dear editor and reviewers,

We thank all of you for your comments and suggestions, which are very helpful for us to improve this manuscript. We carefully revised the manuscript (especially drivers of the trends) according to your comments and suggestions (Please see our point-to-point response below in blue and purple text). We also improved the English with the help of native speakers (see Language certificate). We hope that this revised version could meet the requirements for publishing consideration.

Your sincerely,

All authors

Reviewer #1 (Remarks to the Author):

1. The authors have identified a valid trend in increasing human activity in highlands, this is interesting and significant since the human pressure on highlands can have different and serious impacts on these ecosystems while also contributing to human development. However, it is noteworthy that these trends are not universal as in the case of the Hindu Kush Himalayan region where the population movement to the low lands is still very significant (e.g. as in Ref: P 31 in The Hindu Kush Himalaya Assessment: Mountains, climate change, sustainability and people). To this effect, the questions raised by this paper are noteworthy and contributes significantly to the evidence necessary to account for our response to climate change.

Response: Many thanks for your acknowledgement and positive comments.

2. The authors have provided sufficient evidence through spatial approaches to answer the questions they raised. The methodology is accessible and easily predictable given the data is shared along with the paper.

Response: We appreciate the comment and support from the reviewer.

3. However, it is not clear what is driving these trends especially when developing highlands is still more costly today than ever given many countries are making it difficult to disturb mountain slopes and mountain ecosystems. There is a need to

provide a robust argument for this trend.

Response: Thank you for the important and constructive comments. We analyzed and discussed the driving forces of the trends in a tabular form according to your suggestion (Please see our response to the question of NO.11, and we provided a clear explanation for all the trends in a tabular form).

4. Though sounds trivial, the authors need to clarify the distinction between hills and mountains (the criteria), especially the criteria used to demarcate/identify them in the spatial datasets they used.

Response: We apologize for the insufficient details and thank you for your suggestions for improvement. We improved and clarified the criteria used to demarcate/identify highland (i.e. hills and mountains) according to Digital elevation model (DEM) in Methods part.

Improved some sentences in Methods part (Digital elevation model (DEM) - derived landforms):

We used ASTER GDEM (3935 tiles) to derive the major landforms (i.e. lowland and highland)⁶² in Asia as the base map for subsequent analyses. Lowland includes plains and terraces, while highland refers to the collection of upland (hills) and montane (mountains). The classification criteria of highland and lowland are based on elevation and slope (see details in Supplementary Table 9), which is summarized by Margono and colleagues according to large previous studies⁶². The spatial distributions of lowlands and highlands in Asia were identified with batch processing module of ArcGIS 10.2 platform. Overall, highlands and lowlands are widely distributed in Asia, and their areas accounted for about 40.41% and 59.59%, respectively (Fig. 1).

Supplementary Table 9. Criteria used to determine major landforms based on broad physiographic features (sources, summarized by Margono et al.⁶²).

Categories	Major landforms	Broad physiographic features	Criteria of elevation and slope classes
Lowland	Lowland	Plains and terraces	elevation 0-1000m with slope 0-15° elevation 1000-1200m with slope 0-8°
	Upland	Hills	elevation 0-1000 m with slope >15° elevation 1000-1200 m with slope >8°
Highland	Montane	Mountains	elevation 1200-1500 with slope 0-3° elevation 1200-1500 with slope >3° and elevation >1500m

5. Line 116-119: Many of these countries may have the least amount of lowland to expand into and are largely hilly in nature and hence the expansion in the highlands is the only natural. There appear to be no interesting or contrary observations to be made

here? If some of these countries have plenty of flat lands to expand into, why are they not doing it? It would be interesting to see some arguments on these lines.

Response: Many thanks for your constructive comment and suggestion. We re-analyzed Fig 2(d) through putting percentage of highland areas for each country, and found an interesting (or contrary) observation in one country (Yemen). We improved the original Line 116-119 to present the interesting finding, and added a reasonable explanation of the driving force in Discussion part.

Improved sentences in Results part:

The highland human activity expansion rate in 15 countries exceeded the Asia level (Asia level \approx 23%), and the top 10 countries with the highest highland human activity expansion rates were Bhutan (89.25%), Nepal (75.23%), Armenia (71.12%), Iran (56.85%), Yemen (55.16%), Afghanistan (53.71%), Kyrgyzstan (50.28%), Mongolia (42.13%), Turkey (42.08%) and North Korea (37.61%) (Fig.2d). Most of these countries have very high proportion of highlands, in which Yemen only holds 40.35% highlands but its expansion rate in highlands was more than 55% (Fig. 2d), which is an unusual contrary observation in Yemen compared with other countries.

Added sentences in Discussion part:

Notably, Yemen has the least highland areas (only accounts for 40.35%) among these countries, while the highland developments was $>$ 55% (Fig. 2d), which is mainly due to following reasons: most of the lowlands (e.g coastal plains) are semi-desert, and the highlands are fertile and suitable for developments; and Yemen's civil war exacerbated long-term political instability and economic backwardness, thus Yemen strengthened the developments and utilizations of highlands (including agricultural developments, oil /gas /mineral explorations) to promote livelihoods and economic recovery²⁴⁻²⁶.

Added References:

24. Encyclopedia of the Nations. Yemen country profile. Available at: <https://www.Nation sencyclo pedia.com/Asia-and-Oceania/Yemen.html> (2022)
25. Yadav, S.P. & Lynch, M. Politics, Governance and Reconstruction in Yemen. (Washington, Pomes Studies, 2018).
26. Sowers, J. & Weinthal, E. Humanitarian challenges and the targeting of civilian infrastructure in the Yemen war. *Int. Aff.* **97**, 157-177 (2021).

6. The paper is good in identifying what is happening but lacks strength in providing

an argument on why it is happening. Is it the saturation of lowlands for further expansion, is it the increasing economic and policy ability to occupy highlands more than before? Obviously, these questions are beyond the scope of this paper but are very important offshoots of the questions the authors raised.

Response: Many thanks for your comment and sharing your insights on the topic. We provided more arguments on why it is happening from multifactor perspective, including topographic conditions, marketing demand, population growth/livelihood and dynamic political economy/ecology (dynamic political economy/ecology is also the second reviewer's suggestion; Please see our response to the raised question NO.11, and we provided a clear explanation for all the trends). We also realized the importance of the offshoots proposed in this article, and we added two sentences (hints) of future work in the penultimate paragraph of the Discussion.

Added sentences in Discussion part:

Although Asian highlands hold great potentials for social developments, we should explore the threshold of balancing highland developments and ecological protection in future works. Moreover, considering that highlands are rich in biodiversity, we need to further explore the impacts of highland developments on biodiversity.

7. Though most observations are obvious, the paper identifies the pertinent socio-economic and environmental implications of human expansion into highlands. These observations have important policy implications.

Response: Many thanks for your acknowledgement.

8. Line 157-177: Interesting trends between developing and developed countries in terms of ecological lands being converted to human use. Why developing countries are increasingly relying upon highlands? What is driving them? Policy failures or a conscious decision to engage natural resources for livelihood generation which appears to be the case based on the arguments made in the section starting from L186 but needs to be clearly stated. Do we have evidence supporting these cause-effect relations?

Response: Many thanks for your constructive comments. We provided some evidence in Discussion part (at the beginning of the third paragraph) from the perspective of environmental Kuznets Curve of low income and middle-high income countries. Moreover, our response to the raised question NO.11 also provided some evidences supporting these cause-effect relations).

Added some sentences in Discussion part:

We observed that highland developments caused considerable ecological land loss and fragmentation. Transforming ecological lands into human use in highlands exhibited higher ecological costs in low and lower-middle income countries (~ 1.9 times the rate of ecological land loss) than in high and upper-middle income

countries, which corresponds to the law of environmental Kurtz curve in the process of economic development in developing and developed countries. Previous studies have proved that, for countries just in the stage of economic take-off (e.g. low and lower-middle income countries), reducing poverty and promoting rapid economic growth force these countries to make extensive use of natural endowment to realize industrial take-off, resulting in ecological damage in the early stage of development⁴²⁻⁴⁴. In addition, international trade and investment can lead to the situation of "producing products and consuming natural resources in low-income countries, while enjoy benefits in high-income countries", which improves the ecological environments of developed countries and exacerbates the ecological problems of developing countries⁴⁵. Therefore, the above evidences may well explain the large ecological costs caused by the highland developments in low and low-income countries.

Added References:

42. Kuznets, S. Economic Growth and Income Inequality. *Am. Econ. Rev.* **45**, 1-28 (1955).
43. Mather, A. S., Needle, C. L. & Fairbairn, J. Environmental kuznets curves and forest trends. *Geography* **84**, 55-65 (1999).
44. Rudel, T. K. *et al.* Forest transitions: towards a global understanding of land use change. *Global Environ. Chang.* **15**, 23-31 (2005).
45. López, R, T. M. Economic development and environmental sustainability: new policy options (Oxford Univ. Press, Oxford, 2007).

9. It is good that some reasons for expansion are discussed in L 209-212. We need more such possible reasons to make this paper more usable for policy purposes and by establishing cause-effect relations.

Response: Thank you for acknowledgement and suggestion. We added more possible reasons by establishing cause-effect relations in Discussion part (Please see our response to the raised question NO.8 &11, and we provided more explanations for possible reasons).

10. L 219-224: The question here is why? What is driving these seemingly contrary trends?

Response: Many thanks for your comment. Actually, the population carrying rate in highland expansion areas in most Asian countries is relatively low (much lower than that in lowlands), thus we improved the original sentences to make readers understand it more clearly. We also gave explanations on driving forces (Please see our response to the question NO.11).

Original Line 116-119:

The population capacity rate in the highlands in 16 countries exceeded the Asia level (Fig. 4d, 16 countries inside the purple border), indicating that the role of artificial surfaces in the highlands for supporting population in these countries was

significantly higher than that in other Asian countries. Among these 16 countries, the highland population capacity rate of Mongolia, Bhutan, Nepal and Iran varied from 50% to 78% (Fig. 4d and Supplementary Table 8), denoting that artificial surface development in highlands became the dominant method for improving the population capacity.

After improvement:

The population capacity rate of artificial surface expansion areas in the highlands of 16 Asian countries exceeded the Asia level (Fig. 4d, 16 countries inside the purple border), indicating that the contributions of using highlands to carry population in these countries was significantly higher than those in other countries. Notably, among these 16 countries, the highland population capacity rate of Mongolia, Bhutan, Nepal and Iran varied from 50% to 78% (Fig. 4d and Supplementary Table 8), indicating that their artificial surface developments in the highlands were the dominant method for improving population capacity, considering these 4 countries all have very high highland proportions and highland development rates (Fig. 2d and Supplementary Fig. 8a).

11. The text from Line 243 provides some good explanations for trends but not for all the regions, it would be good to provide a clear explanation for all the trends in a tabular form, for e.g.

Response: Many thanks for your suggestion. We improved this part through literature review, expert consultation and referees' suggestions (from more dynamic political economy/ ecology analysis). The revised details are shown in second paragraph and Table 1 of Discussion part.

Improved sentences and added Table 1 in Discussion:

We analysed the dominant and potential drivers affecting the human activity expansions in the highlands in different Asian countries during 2000 to 2020 from multifactor perspective (Table 1). Generally, for the countries with extremely high ($\approx 90\%$) or very low human activity expansion rates in the highlands ($< 2\%$) (only one-fifth countries), topographic constraint may be the dominant factor affecting highland developments, considering that these countries possess extremely high or low proportion of highlands (Table 1). However, other four-fifths Asian countries had 5%–95% highland areas (Supplementary Fig 12), and thus explaining the human activity expansion rates in the highlands in these countries using topographic factors is unfair. Highland developments may be driven by different needs, and the superposition of multiple factors could result in the highland developments with

different degrees in these countries, including topographic condition, marketing demand, population growth/livelihood and dynamic socio-political economy/ecology factors (details in Table 1).

Table 1 Dominant and potential drivers of human activity expansions in the highlands in different Asian countries during 2000 to 2020.

Degrees of human activity expansion	Countries	Dominant and potential drivers	
Extremely high ($\approx 90\%$) or very low human activity expansion rates in the highlands ($< 2\%$)	Bhutan, Qatar, Bahrain, Bangladesh, Maldives, Iraq, Turkmenistan, Kuwait, United Arab Emirates	In Bhutan, flat land resources are scarce and more than 95% of its land is highlands (Fig.2d). Thus, highland development is obligatory for maintaining social and economic developments. Such situation explains the extremely high human activity expansion rate in the highlands in Bhutan ($\approx 90\%$). In contrast, the scarce highlands in some countries (highland proportion $< 5\%$ in Qatar, Bahrain, Bangladesh, Maldives, Iraq, Turkmenistan, Kuwait and United Arab Emirates, Supplementary Fig.12a) can explain their low human activity expansion rates in the highlands ($< 2\%$). Therefore, topographic factors significantly impact highland developments in the abovementioned countries ($R^2 = 0.9981$, $p < 0.01$, Supplementary Fig 12b).	
Very high human activity expansion rate in the highlands ($\sim 38\%$ - 76%)	Nepal, Armenia, Iran, Yemen, Afghanistan, Kyrgyzstan, Mongolia, Turkey, North Korea	Nepal, Armenia, Iran, Afghanistan, Kyrgyzstan, Mongolia, Turkey and North Korea have high highland rates (proportion of highlands $> 57\%$, Fig. 2d), therefore the human activity expansions in the highlands are natural phenomenon. Since 2006, a nationwide “Development of Pastures and Meadows and Pasture and Forage Crop Production Project” in Turkey contributed to the agricultural developments in the highlands ²² . Mongolia’s western, northern and northeastern parts are upland and suitable for farming, and a national project for the development of farming was started since 1997 to improve food security and increase family incomes to alleviate poverty, which has largely driven Mongolia's highland developments ²³ . Notably, Yemen has the least highland areas (only accounts for 40.35%) among these countries, while the highland developments was $> 55\%$ (Fig. 2d), which is mainly due to following reasons: most of the lowlands (e.g coastal plains) are semi-desert, and the highlands are fertile and suitable for developments; and Yemen’s civil war exacerbated long-term political instability and economic backwardness, thus Yemen strengthened the developments and utilizations of highlands (including agricultural developments, oil /gas /mineral explorations) to promote livelihoods and economic recovery ²⁴⁻²⁶ .	Highlands are rich in agro-biodiversity and genetic resources, and Central and West Asia countries (i.e. Afghanistan, Iran, Tajikistan ,Turkey, Pakistan, Azerbaijan and Georgia,) have built collaboration on rainfed agricultures in highlands with the International Center for Agricultural Research in the Dry Areas to support growing populations and reduce poverty ²² . The transitional economy for Central Asian countries and North Korea (i.e. transformation from planned economy to market economy or mixed economy) have boosted the developments of infrastructure, industry and agriculture, while strengthening the utilizations of resources and export trade ²⁷ , which may lead to the widespread highland developments of Kyrgyzstan, Tajikistan, Pakistan, Uzbekistan Kazakhstan and North Korea.
High human activity expansion rate in the highlands ($> \text{Asia level}$) ($\sim 23\%$ - 34%)	Lebanon, Tajikistan, China, Vietnam, Georgia	Lebanon faces water stress and food insecurity, while its highlands have abundant rainfall (~ 1400 mm average annual rainfall), which drive Lebanon to develop highlands to a certain extent ³⁴ .The ‘Development of the Western Region’ policy (comprising 15 mountainous provinces and regions in the central and	Marketing demands (e.g. timber export, expansions of cash crops-corn, tea, coffee and upland rice), land scarcity (by

		western China) issued in 2000 and the ‘Low-slope Hilly Regions Comprehensive Development and Utilization’ policy formulated in 2006 considerably contributed to the human activity expansion rate in the highlands in China ^{7,35} .	2000, most of the mainland Southeast Asian’s lowlands were used for some form of agriculture) and national land-tenure policies intensified the expansions of cultivated lands and artificial surfaces in the highlands of Southeast Asian countries (e.g. Vietnam, Malaysia, Indonesia, Philippines, Thailand, Laos, East Timor, Brunei and Cambodia) ^{13,36-39} .	from lowlands to highlands, away from areas vulnerable to drought and flooding from sea level rise) drove highland developments in these countries ²⁸ . Since the 21st century, the global urbanization has been accelerating significantly (urban population has increased from 30% in 1950 to 56% in 2019), especially in Asia and Africa ²⁹ . To meet the needs of urban population growth and rapid urbanization as well as people’s pursuit of fresh air, landscape, exclusivity and closeness to nature, the hillsides around urban regions are being developed at an accelerating rate in different Asian cities ^{7,15,30,31} , which drove the highland developments in many Asian countries. China’s “One Belt One Road” initiative increased direct investments and the construction of various infrastructures in Asian countries (e.g. railways, highways, hydropower stations, natural gas pipeline projects), which contributed to the highland developments in Asian countries to a certain extent ³² . Highlands have beautiful natural landscapes, and Asian countries have accelerated the developments of mountain tourisms in the 21st century, which also drove highland developments to a certain extent ³³ .
Medium human activity expansion rate in the highlands (<Asia level) (~5% -20%)	Pakistan, Laos, Cyprus, Japan, South Korea, Myanmar, Philippines, Palestine, Indonesia, Jordan, Thailand, Saudi Arabia, Malaysia, Syria, Azerbaijan, Brunei, India, Uzbekistan, East Timor	In order to maintain/protect mountain agriculture, Japan promulgated the Depopulated Areas Emergency Act and the Mountain Villages Development Act, which promoted the developments of highlands in Japan ²³ . Highland farming was expanded to produce commercial crops and has become a major income source for farm households, which can drive highland developments in South Korea ²³ . Since 2004, the Annan peace plan led to a construction boom in Cyprus, and then induced disorderly land developments ⁴⁰ . After joining the World Trade Organization in 1999, Jordan strengthened economic regulation and control, and took corresponding measures in finance, infrastructure, investment attraction and foreign aid ⁴¹ , which may drive the developments of highlands. India adopted watershed approach based on the principle of people’s participation to develop the uplands in Indian Himalayas and south India ²³ .		
Low human activity expansion rate in the highlands (<Asia level) (~2% -5%)	Israel, Singapore, Cambodia, Sri Lanka, Kazakhstan	These countries have relatively scarce highlands (Fig. 2d), therefore most human activities are concentrated in lowlands.		

Improved and Added References:

15. Yang, C. *et al.* Characteristics and trends of hillside urbanization in China from 2007 to 2017. *Habitat Int.* **120**, 102502 (2022).
22. Roozitalab, M. H., Serghini, H., Keshavarz, A., Eser, V., & Depauw, E. Sustainable Agricultural Development of Highlands in Central, West Asia, and North Africa. Synthesis of Regional Expert Meeting on Highland Agriculture November 2011, Karaj, Iran (2013).
23. Partap, T., & Chancellor, V. Sustainable farming systems in upland areas. Report of the APO study meeting on sustainable farming systems in upland areas, New Delhi, 15–19 Jan 2001. Asian Productivity Organization, Tokyo (2004).
24. Encyclopedia of the Nations. Yemen country profile. Available at:

- <https://www.nationsencyclopedia.com/Asia-and-Oceania/Yemen.html> (2022)
25. Yadav, S.P. & Lynch, M. *Politics, Governance and Reconstruction in Yemen*. (Washington, Pomes Studies, 2018).
 26. Sowers, J. & Weinthal, E. Humanitarian challenges and the targeting of civilian infrastructure in the Yemen war. *Int. Aff.* **97**, 157-177 (2021).
 27. Park, J., Kang, B., Min, J., Gwun, K. & Yun, C. Economic Development Strategies of Major Central Asian Countries and Their Implications for Korea. KIEP Research Paper, World Economy Brief. Available at: <https://ssrn.com/abstract=3089482> (2017).
 28. Sutton, W. R., Srivastava, J. P., Neumann, J.E., Strzpek, K. M. & Droogers P. Reducing the vulnerability of Azerbaijan's agricultural systems to climate change: impact assessment and adaptation options (World Bank Publications, Washington, 2013).
 29. World bank. World bank database. Available at: <https://data.worldbank.org.cn/> (2020).
 30. Too, E. G., Adnan, N. & Trigunarsyah, B. Project governance in Malaysia hillside developments. 6th International Conference on Construction in the 21st Century (CITC-VI), Kuala Lumpur Malaysia. (2011).
 31. Ahn, J. E. *Cities on the Edge: Significance and Preservation of Hillside Squatter Settlements in Korea*. (Columbia University, New York, 2014).
 32. Foo, N., Lean, H. H. & Salim, R. The impact of China's One Belt One Road initiative on international trade in the ASEAN region. *N. Am. J. Econ. Financ.* **54**, 101089 (2019).
 33. Bui, H. T., Jones, T.E. & Apollo, M. *Nature-Based Tourism in Asia's Mountainous Protected Areas* (Geographies of Tourism and Global Change, Springer, Cham, 2021).
 34. Kallas, G., Palacios-Rodriguez, G. & Kattar, S. Land Suitability for Biological Wastewater Treatment in Lebanon and the Litani River Basin Using Fuzzy Logic and Analytical Hierarchy Process. *Forests* **13**, 139 (2022).
 40. Yorucu, V., & Keles, R. The construction boom and environmental protection in northern Cyprus as a consequence of the Annan plan. *Constr. Manag. Econ.* **25**, 77-86 (2007).
 41. MOFA. *The Hashemite Kingdom of Jordan*. Available at: <http://www.fmprc.gov.cn/> (2021).
-

Reviewer #2 (Remarks to the Author):

1. Mountains provide 50% of global fresh water, offer habitat for some 33% of terrestrial biodiversity, harbor half of all global biodiversity hotspots (Rumbine & Xu 2021, *Front Ecol Environ*). Asian highlands are the most dynamic, diverse and complex landscapes at worldwide. This paper provides timely insights for land expansion in the Asian highlands. However it needs to well definitely defined highland boundary. The commonly used definition is from WCMC-UNEP 2002 and Kapos et al. 2000. According to their definition, only around 1/4 of world terrestrial ecosystem is called mountains or hills. It would be nice to tell the difference between hills and mountains for readers.

Response: Many thanks for your acknowledgement and suggestion. We put the information provided by Rumbine & Xu 2021 in the Introduction part to further highlight the significance of our research.

Improved sentence in Introduction:

The geographically diverse highlands are ecologically fragile regions, which play important roles in biodiversity conservation (offering habitat for ~33% of terrestrial biodiversity), carbon sequestration, water supply (over 50% of global fresh water) and soil and water conservation⁸⁻¹².

Added References:

12. Grumbine, R. E. & Xu, J. C. Mountain futures: pursuing innovative adaptations in coupled social-ecological systems. *Front Ecol. Environ.* **19**, 342-348 (2021).

We did carefully consider the definition of highland boundary, including WCMC-UNEP 2002 and Kapos et al. 2000. The reasons for us to finally apply the latest standard (Margono et al, *Nat. Clim. Change*, 2014) are: (1) we obtained the highland boundaries in Asian with above two standards (as shown in the figure below). Asian highland proportion is 40.41% for the standard applied by this study, while 35.30% for WCMC. There is a little difference between the two results. Considering the large area of China in Asia, we calculated the proportion of China's highlands considering the two standards (as shown in the figure below). According to China's survey and common standards, China's highlands are more than 2/3 (~69%), and the result from the standard we used is closer to the China's situation. (2) WCMC did not give a detailed numerical standard for dividing mountains and hills in highlands, while the standard we used presents detailed values (Supplementary Table 9). Nevertheless, we also added a sentence in Discussion to discuss this issue.

Asia highlands (defined as hilly and mountainous regions) account for 40.41% in our study; China highlands account for 62.47% of China's land area.

Asia highlands (defined as mountains or mountain areas) account for 35.30% in WCMC-UNEP and Kapos et al. 2000; China highlands account for 52.76% of China's land area.

Added sentence in Discussion:

Since no unified standard for the division of highlands exists, we used the latest standards (Supplementary Table 9); however, other standards (e.g. World Conservation Monitoring Centre-UN Environment Program (WCMC-UNEP)⁵⁸) may also hold potentials in mapping Asian highlands.

58. Kapos, V. R. J., Edwards, M., Price, M.F., & Ravilious, C. Developing a map of the world's mountain forests. In: Price, M.F, Butt, N.(eds) Forests in sustainable mountain development: a report for 2000. (CAB International, Wallingford, 2000).

2. I like to the statement of "Highland development supports sustainable development in Asia", but authors have failed to provide alternatives on how to do it. Balmford et al. (2016) talked "Getting Road Expansion on Right Track" as a good example. His team is also developed "land sharing" and "land sparing" approach for agricultural expansion.

Response: Many thanks for your comments. We added one paragraph in Discussion part to provide alternatives on how to do it.

Added paragraph in Discussion:

Although Asian highland developments prevented the net loss of total cultivated lands and provided living and working spaces for the growing population, considering the negative ecological impacts of highland developments, Asian countries should balance highland developments and ecological protection for their sustainable developments, rather than blind developing. Previous studies have proved that "land sharing" (i.e. integrating biodiversity conservation and food production on the same land with wildlife-friendly farming methods) and "land sparing" (i.e. separating land for conservation from land for crops, with high-yield farming facilitating the protection of remaining natural habitats from agricultural expansion) could provide alternatives on supporting agricultural sustainable developments while mitigating negative effects⁵³⁻⁵⁵. Meanwhile, "getting road expansion on right track" (i.e. improving and planning transport links to substantially increase food production at relatively limited environmental cost) could enhance livelihoods and regional food production and reduce overexploitation of cultivated land while helping safeguard vital ecosystem services and significant biological diversity⁵⁶. Therefore, integrating the strategies of "land sharing", "land sparing" and planning transport links in Asian highland developments may help save land resources and protect the ecological environment in highlands, and make highland developments achieve sustainability.

Added references:

- 53 Phalan, B., Onial, M., Balmford, A. & Green, R. E. Reconciling Food Production and Biodiversity Conservation: Land Sharing and Land Sparing Compared. *Science* **333**, 1289-1291(2011).
- 54 Green, R. E., Cornell, S. J., Scharlemann, J. P. W. & Balmford, A. Farming and the fate of wild nature. *Science* **307**, 550-555(2005).
- 55 Balmford, A., Green, R. E. & Scharlemann, J. P. W. Sparing land for nature: exploring the potential impact of changes in agricultural yield on the area needed for crop production. *Global Change Biol.* **11**, 1594-1605 (2005).
- 56 Balmford, A. et al. Getting Road Expansion on the Right Track: A Framework for Smart Infrastructure Planning in the Mekong. *Plos Biol.* **14**, e2000266 (2016).

3. For drivers of land expansion in highlands, instead of stereotype socioeconomic data, the paper could be benefit from more dynamic political economy/ecology analysis, such as environmental Kuznets Curve for low income and middle-high income countries (Rudel at al. *Global Environmental Change* 15(1):23-21), transitional economy for Central Asian countries and North Korea, civil wars for Yeman and Afghanistan. The English needs significant improvement too.

Response: Thank you for your constructive comments. Following your suggestion, we improved drivers of land expansion in highlands considering topographic conditions, marketing demand, population growth/livelihood and dynamic socio-political economy/ecology factors. We also improved the English with the help of native speakers (see Language certificate).

Improved sentences and added Table in Discussion part:

We analysed the dominant and potential drivers affecting the human activity expansions in the highlands in different Asian countries during 2000 to 2020 from multifactor perspective (Table 1). Generally, for the countries with extremely high ($\approx 90\%$) or very low human activity expansion rates in the highlands ($< 2\%$) (only one-fifth countries), topographic constraint may be the dominant factor affecting highland developments, considering that these countries possess extremely high or low proportion of highlands (Table 1). However, other four-fifths Asian countries had 5%–95% highland areas (Supplementary Fig 12), and thus explaining the human activity expansion rates in the highlands in these countries using topographic factors is unfair. Highland developments may be driven by different needs, and the superposition of multiple factors could result in the highland developments with different degrees in these countries, including topographic condition, marketing demand, population growth/livelihood and dynamic socio-political economy/ecology factors (details in Table 1).

Table 1 Dominant and potential drivers of human activity expansions in the highlands in different Asian countries during 2000 to 2020.

Degrees of human activity expansion	Countries	Dominant and potential drivers	
Extremely high ($\approx 90\%$) or very low human activity expansion rates in the highlands ($< 2\%$)	Bhutan, Qatar, Bahrain, Bangladesh, Maldives, Iraq, Turkmenistan, Kuwait, United Arab Emirates	In Bhutan, flat land resources are scarce and more than 95% of its land is highlands (Fig.2d). Thus, highland development is obligatory for maintaining social and economic developments. Such situation explains the extremely high human activity expansion rate in the highlands in Bhutan ($\approx 90\%$). In contrast, the scarce highlands in some countries (highland proportion $< 5\%$ in Qatar, Bahrain, Bangladesh, Maldives, Iraq, Turkmenistan, Kuwait and United Arab Emirates, Supplementary Fig.12a) can explain their low human activity expansion rates in the highlands ($< 2\%$). Therefore, topographic factors significantly impact highland developments in the abovementioned countries ($R^2 = 0.9981, p < 0.01$, Supplementary Fig 12b).	
Very high human activity expansion rate in the	Nepal, Armenia, Iran, Yemen, Afghanistan,	Nepal, Armenia, Iran, Afghanistan, Kyrgyzstan, Mongolia, Turkey and North Korea have high highland rates (proportion of highlands $> 57\%$, Fig. 2d), therefore the human activity expansions in the highlands are natural phenomenon. Since	Highlands are rich in agro-biodiversity and genetic resources, and Central and West Asia countries (i.e. Afghanistan, Iran, Tajikistan ,Turkey,

highlands (~38% -76%)	Kyrgyzstan, Mongolia, Turkey, North Korea	2006, a nationwide “Development of Pastures and Meadows and Pasture and Forage Crop Production Project” in Turkey contributed to the agricultural developments in the highlands²². Mongolia’s western, northern and northeastern parts are upland and suitable for farming, and a national project for the development of farming was started since 1997 to improve food security and increase family incomes to alleviate poverty, which has largely driven Mongolia’s highland developments²³. Notably, Yemen has the least highland areas (only accounts for 40.35%) among these countries, while the highland developments was > 55% (Fig. 2d), which is mainly due to following reasons: most of the lowlands (e.g. coastal plains) are semi-desert, and the highlands are fertile and suitable for developments; and Yemen’s civil war exacerbated long-term political instability and economic backwardness, thus Yemen strengthened the developments and utilizations of highlands (including agricultural developments, oil /gas /mineral explorations) to promote livelihoods and economic recovery²⁴⁻²⁶.		Pakistan, Azerbaijan and Georgia.) have built collaboration on rainfed agricultures in highlands with the International Center for Agricultural Research in the Dry Areas to support growing populations and reduce poverty²². The transitional economy for Central Asian countries and North Korea (i.e. transformation from planned economy to market economy or mixed economy) have boosted the developments of infrastructure, industry and agriculture, while strengthening the utilizations of resources and export trade²⁷, which may lead to the widespread highland developments of Kyrgyzstan, Tajikistan, Pakistan, Uzbekistan Kazakhstan and North Korea.
High human activity expansion rate in the highlands (>Asia level) (~23% -34%)	Lebanon, Tajikistan, China, Vietnam, Georgia	Lebanon faces water stress and food insecurity, while its highlands have abundant rainfall (~1400 mm average annual rainfall), which drive Lebanon to develop highlands to a certain extent³⁴. The ‘Development of the Western Region’ policy (comprising 15 mountainous provinces and regions in the central and western China) issued in 2000 and the ‘Low-slope Hilly Regions Comprehensive Development and Utilization’ policy formulated in 2006 considerably contributed to the human activity expansion rate in the highlands in China^{7,35}.	Marketing demands (e.g. timber export, expansions of cash crops-corn, tea, coffee and upland rice), land scarcity (by 2000, most of the mainland Southeast Asian’s lowlands were used for some form of agriculture) and national land-tenure policies intensified the expansions of cultivated lands and artificial surfaces in the highlands of Southeast Asian countries (e.g. Vietnam, Malaysia, Indonesia, Philippines, Thailand, Laos, East Timor, Brunei and Cambodia)^{13,36-39}.	From 2009, Armenia, Azerbaijan and Georgia participated in the part of the World Bank’s Europe and Central Asia (ECA) Regional Analytical and Advisory Activities Program on Reducing Vulnerability to Climate Change in ECA Agricultural Systems program, of which more systematic land management (i.e. shifting crops from lowlands to highlands, away from areas vulnerable to drought and flooding from sea level rise) drove highland developments in these countries²⁸.
Medium human activity expansion rate in the highlands (<Asia level) (~5% -20%)	Pakistan, Laos, Cyprus, Japan, South Korea, Myanmar, Philippines, Palestine, Indonesia, Jordan, Thailand, Saudi Arabia, Malaysia, Syria, Azerbaijan, Brunei, India, Uzbekistan, East Timor	In order to maintain/protect mountain agriculture, Japan promulgated the Depopulated Areas Emergency Act and the Mountain Villages Development Act, which promoted the developments of highlands in Japan²³. Highland farming was expanded to produce commercial crops and has become a major income source for farm households, which can drive highland developments in South Korea²³. Since 2004, the Annan peace plan led to a construction boom in Cyprus, and then induced disorderly land developments⁴⁰. After joining the World Trade Organization in 1999, Jordan strengthened economic regulation and control, and took corresponding measures in finance, infrastructure, investment attraction and foreign aid⁴¹, which may drive		Since the 21st century, the global urbanization has been accelerating significantly (urban population has increased from 30% in 1950 to 56% in 2019), especially in Asia and Africa²⁹. To meet the needs of urban population growth and rapid urbanization as well as people’s pursuit of fresh air, landscape, exclusivity and closeness to nature, the hillsides around urban regions are being developed at an accelerating rate in different Asian cities^{7,15,30,31}, which drove the highland developments in many Asian countries. China’s “One Belt One Road” initiative increased direct investments and the construction of various infrastructures in Asian countries (e.g. railways,

		the developments of highlands. India adopted watershed approach based on the principle of people's participation to develop the uplands in Indian Himalayas and south India ²³ .		highways, hydropower stations, natural gas pipeline projects), which contributed to the highland developments in Asian countries to a certain extent ³² .
Low human activity expansion rate in the highlands (<Asia level) (~2%-5%)	Israel, Singapore, Cambodia, Sri Lanka, Kazakhstan	These countries have relatively scarce highlands (Fig. 2d), therefore most human activities are concentrated in lowlands.		Highlands have beautiful natural landscapes, and Asian countries have accelerated the developments of mountain tourisms in the 21st century, which also drove highland developments to a certain extent ³³ .

Added some sentences in Discussion part:

We observed that highland developments caused considerable ecological land loss and fragmentation. Transforming ecological lands into human use in highlands exhibited higher ecological costs in low and lower-middle income countries (~ 1.9 times the rate of ecological land loss) than in high and upper-middle income countries, which corresponds to the law of environmental Kurtz curve in the process of economic development in developing and developed countries. Previous studies have proved that, for countries just in the stage of economic take-off (e.g. low and lower-middle income countries), reducing poverty and promoting rapid economic growth force these countries to make extensive use of natural endowment to realize industrial take-off, resulting in ecological damage in the early stage of development⁴²⁻⁴⁴. In addition, international trade and investment can lead to the situation of "producing products and consuming natural resources in low-income countries, while enjoy benefits in high-income countries", which improves the ecological environments of developed countries and exacerbates the ecological problems of developing countries⁴⁵. Therefore, the above evidences may well explain the large ecological costs caused by the highland developments in low and low-income countries.

Improved and Added References:

15. Yang, C. *et al.* Characteristics and trends of hillside urbanization in China from 2007 to 2017. *Habitat Int.* **120**, 102502 (2022).
22. Roozitalab, M. H., Serghini, H., Keshavarz, A., Eser, V., & Depauw, E. Sustainable Agricultural Development of Highlands in Central, West Asia, and North Africa. Synthesis of Regional Expert Meeting on Highland Agriculture November 2011, Karaj, Iran (2013).
23. Partap, T., & Chancellor, V. Sustainable farming systems in upland areas. Report of the APO study meeting on sustainable farming systems in upland areas, New Delhi, 15–19 Jan 2001. Asian Productivity Organization, Tokyo (2004).
24. Encyclopedia of the Nations. Yemen country profile. Available at: <https://www.nationsencyclopedia.com/Asia-and-Oceania/Yemen.html> (2022)
25. Yadav, S.P. & Lynch, M. Politics, Governance and Reconstruction in Yemen. (Washington, Pomes Studies, 2018).
26. Sowers, J. & Weinthal, E. Humanitarian challenges and the targeting of civilian infrastructure in

- the Yemen war. *Int. Aff.* **97**, 157-177 (2021).
27. Park, J., Kang, B., Min, J., Gwun, K. & Yun, C. Economic Development Strategies of Major Central Asian Countries and Their Implications for Korea. KIEP Research Paper, World Economy Brief. Available at: <https://ssrn.com/abstract=3089482> (2017).
 28. Sutton, W. R., Srivastava, J. P., Neumann, J.E., Strzpek, K. M. & Droogers P. Reducing the vulnerability of Azerbaijan's agricultural systems to climate change: impact assessment and adaptation options (World Bank Publications, Washington, 2013).
 29. World bank. World bank database. Available at: <https://data.worldbank.org.cn/> (2020).
 30. Too, E. G., Adnan, N. & Trigunaryah, B. Project governance in Malaysia hillside developments. 6th International Conference on Construction in the 21st Century (CITC-VI), Kuala Lumpur Malaysia. (2011).
 31. Ahn, J. E. Cities on the Edge: Significance and Preservation of Hillside Squatter Settlements in Korea. (Columbia University, New York, 2014).
 32. Foo, N., Lean, H. H. & Salim, R. The impact of China's One Belt One Road initiative on international trade in the ASEAN region. *N. Am. J. Econ. Financ.* **54**, 101089 (2019).
 33. Bui, H. T., Jones, T.E. & Apollo, M. Nature-Based Tourism in Asia's Mountainous Protected Areas (Geographies of Tourism and Global Change, Springer, Cham, 2021).
 34. Kallas, G., Palacios-Rodriguez, G. & Kattar, S. Land Suitability for Biological Wastewater Treatment in Lebanon and the Litani River Basin Using Fuzzy Logic and Analytical Hierarchy Process. *Forests* **13**, 139 (2022).
 40. Yorucu, V., & Keles, R. The construction boom and environmental protection in northern Cyprus as a consequence of the Annan plan. *Constr. Manag. Econ.* **25**, 77-86 (2007).
 41. MOFA. The Hashemite Kingdom of Jordan. Available at: <http://www.fmprc.gov.cn/> (2021).
 42. Kuznets, S. Economic Growth and Income Inequality. *Am. Econ. Rev.* **45**, 1-28 (1955).
 43. Mather, A. S., Needle, C. L. & Fairbairn, J. Environmental kuznets curves and forest trends. *Geography* **84**, 55-65 (1999).
 44. Rudel, T. K. *et al.* Forest transitions: towards a global understanding of land use change. *Global Environ. Chang.* **15**, 23-31 (2005).
 45. López, R, T. M. Economic development and environmental sustainability: new policy options (Oxford Univ. Press, Oxford, 2007).

Language certificate

[REDACTED]

4. More specially:

1) Fig 2(d) top 10 countries are all mountainous countries, please try to put percentage of mountainous areas for each country;

Response: Many thanks for your helpful comment. We put percentage value of highland areas for each country in Fig 2(d).

2) Line 585: "Tailand" should be "Thailand"

Response: We apologize for the confusion. We corrected it.